# Lactoferrin as Possible Treatment for Chronic Gastrointestinal Symptoms in Children with Long COVID: Case Series and Literature Review

**DOI:** 10.3390/children9101446

**Published:** 2022-09-22

**Authors:** Rosa Morello, Cristina De Rose, Sara Cardinali, Piero Valentini, Danilo Buonsenso

**Affiliations:** 1Department of Woman and Child Health and Public Health, Fondazione Policlinico Universitario “Agostino Gemelli”, IRCCS, Università Cattolica Sacro Cuore, 00168 Rome, Italy; 2Department of Laboratory and Infectious Sciences, Università del Sacro Cuore, 20123 Rome, Italy

**Keywords:** SARS-CoV-2 infection, long COVID, children, oral supplementation, lactoferrin

## Abstract

Long COVID is an emergent, heterogeneous, and multisystemic condition with an increasingly important impact also on the pediatric population. Among long COVID symptoms, patients can experience chronic gastrointestinal symptoms such as abdominal pain, constipation, diarrhea, vomiting, nausea, and dysphagia. Although there is no standard, agreed, and optimal diagnostic approach or treatment of long COVID in children, recently compounds containing multiple micronutrients and lactoferrin have been proposed as a possible treatment strategy, due to the long-standing experience gained from other gastrointestinal conditions. In particular, lactoferrin is a pleiotropic glycoprotein with antioxidant, anti-inflammatory, antithrombotic, and immunomodulatory activities. Moreover, it seems to have several physiological functions to protect the gastrointestinal tract. In this regard, we described the resolution of symptoms after the start of therapy with high doses of oral lactoferrin in two patients referred to our post-COVID pediatric unit due to chronic gastrointestinal symptoms after SARS-CoV-2 infection.

## 1. Introduction

Since its discovery, the SARS-CoV-2 virus has had a great impact on people’s health, including children [1]. Although acute infection in children usually has a mild course, other serious consequences, such as multisystem inflammatory syndrome (MISC) and long COVID, should be considered [2,3].

Long COVID is an emergent syndrome of global health concern. It is a systemic condition characterized by the persistence of signs and symptoms for weeks after SARS-CoV-2 infection [4]. About its definition, long COVID syndrome has not yet been clearly defined. In October 2021, the World Health Organization (WHO) proposed the first clinical definition for post-COVID-19 condition [5]. Other possible definitions have been proposed by the National Institute for Health and Care Excellence (on February 2022) and by the National Institutes of Health [4,5]. Stephenson T. et al. proposed the following pediatric long COVID definition: “Post-COVID-19 condition occurs in young people with a history of confirmed SARS-CoV-2 infection, with at least one persisting physical symptom for a minimum duration of 12 weeks after initial testing that cannot be explained by an alternative diagnosis. The symptoms have an impact on everyday functioning, may continue or develop after COVID infection, and may fluctuate or relapse over time” [6]. In this regard, it is difficult to accurately provide the long COVID syndrome incidence and prevalence without a precise and universally recognized clinical case definition. A recent analysis reported a prevalence of 25.24% in the pediatric population and adolescents [4]. In terms of clinical presentation, signs and symptoms from various organs can characterize pediatric long COVID syndrome: respiratory, cardiological, musculoskeletal, neurological, and gastrointestinal symptoms; headache; persistent or recurrent skin lesions; alopecia; sleep and concentration disorders [7]. Gastrointestinal symptoms (abdominal pain, constipation, diarrhea, vomiting/nausea, dysphagia) have a prevalence of less than 5%; chronic diarrhea in particular appears to have a prevalence of 1.68% [4].

The exact etiology of long COVID is still unknown. However, it could be the consequence of multiple pathological mechanisms, including chronic inflammation, immune dysregulation/autoimmunity, the persistence of viral particles, alteration of the coagulation profile, and subsequent chronic endothelial damage [3,8,9,10,11].

Currently, there is no standard, agreed, and optimal diagnostic approach or treatment of long COVID in children, although initial guidance, local recommendations, and consensus are available in literature [3,12]. So far, microelements, vitamins, statins, anticoagulants/anti-aggregants are being variously studied by different research centers [3].

About treatment, considering the potential pathophysiology of long COVID, multivitamin supplements and lactoferrin have recently been proposed as a possible treatment strategy [3,9].

In this regard, we described two pediatric case in which persistent gastrointestinal symptoms after SARS-CoV-2 infection disappeared after the start of therapy with high doses of oral lactoferrin.

## 2. Cases Presentation

### 2.1. Methods

This is a retrospective case series of two children with persistent gastrointestinal symptoms after previous microbiologically-confirmed SARS-CoV-2 infection, evaluated in the pediatric post-COVID outpatient unit of Gemelli University Hospital in Rome, Italy. In our outpatient clinic, we evaluate children that had fully recovered from acute infection and those that presented with persisting symptoms since mid-2020. Children can be sent to the post-COVID unit either after discharge from our institution, or directly sent from the family pediatricians (and therefore not seen at baseline during acute infection). We developed a protocol to offer a personalized assessment of children with long COVID-19, which has been described previously [3]. Therefore, the following categories of children are routinely evaluated in our setting:-Fully recovered children: This group included those that reported no persisting symptoms after acute SARS-CoV-2 infection at the time of follow-up post-onset of acute COVID-19 symptoms (at least 8 weeks).-Long COVID group: any child with persisting symptoms for at least 12 weeks after SARS-CoV-2 infection, that cannot be explained by an alternative diagnosis and have a negative impact on daily life.

Since there are no available or approved treatments for children with long COVID, here we report two patients with unexplained chronic gastrointestinal symptoms successfully treated with Lactoferrin (LF). These patients are part of a retrospective observational study currently under analysis of children with long COVID treated with LF.

### 2.2. Case One

A six-year-old boy was referred to our post-COVID pediatric unit in a third-level hospital for persistent symptoms after COVID-19 disease. He contracted a SARS-CoV-2 infection in December 2021. The acute phase was characterized by fever for two days, headaches, and gastrointestinal symptoms with diarrhea. The child became negative after 10 days. However, after the negativization, diarrhea and asthenia persisted. Then, after 16 weeks from the acute infection, he was led to our clinic. At the evaluation, physical examination and vital signs were normal. There was nothing to report in his past medical history, except for COVID infection. The bedside thoracic and abdomen ultrasound were normal. The blood tests (blood count, blood chemistry, liver and kidney function) were normal; screening for coeliac disease was negative. Chemical and microbiological tests (including parasitological examination) on stool samples were always negative. A fecal SARS-CoV-2 PCR test was performed and resulted positive. Therefore, in the suspicion of long COVID syndrome with predominant gastrointestinal symptoms, we prescribed treatment with oral lactoferrin (600 mg/die) for 90 days. The treatment schedule was based according to the safety of previous similar studies [13]. This lactoferrin formulation contains no other nutrient, neither vitamin c, nor zinc nor others. The complete composition is: lactoferrin; corn maltodextrin; anticaking agents (magnesium salts of fatty acids, silicon dioxide.); capsule in hydrossypropilmethilcellulosa. Parents were instructed to open the capsules and melt the powder in a small amount of water or milk. After the start of therapy, there was a gradual improvement, and after two weeks the therapy led to a complete resolution with the disappearance of diarrhea, while fatigue improved more gradually in two months. SARS-CoV-2 PCR on stool turned negative when tested one month later. In August, the child and his whole family (father, mother, and little brother) had another SARS-CoV-2 infection (Omicron 5). The six-year-old boy was still in therapy with lactoferrin, and he was the only member of the family to not suffer from any symptoms. The parents and the one-year-old brother suffered from fever for about 3 days and other physical diseases such as headaches, asthenia and gastrointestinal symptoms with diarrhea.

### 2.3. Case Two

An eleven-month-old girl was referred to our post-COVID pediatric unit in a third-level hospital for gastrointestinal persistent symptoms after COVID-19 disease. She had SARS-CoV-2 infection in July 2022. The acute phase was characterized by fever for three days. After a few days, due to the diagnosis of urinary tract infection she was prescribed oral antibiotic therapy for 7 days. The infant became negative after 7 days. However, after the negativization, diarrhea persisted. In particular, the mother complained of numerous (about five) liquid stool evacuations every day. Accordingly, in suspicion of lactose intolerance/allergy, the mother had initially eliminated from the baby’s diet milk and derivatives. In the absence of benefit, and in the suspicion of celiac disease, also gluten was subsequently eliminated from the baby’s diet by the autonomous decision of the mother. Not even the therapy with lactic ferments had led to an improvement in symptomatology. Then, in view of the persistence of symptoms for more than 4 weeks, the baby was led to our clinic. At our evaluation, the baby was on a diet free of milk and dairy products, gluten and fiber. Physical examination and vital signs were normal. There was nothing to report in past medical history. The bedside thoracic ultrasound was normal. An initial abdomen ultrasound examination documented mesenteric adenitis. The control ultrasound carried out after about 2 weeks was negative. The blood tests (blood count, blood chemistry, liver and kidney function) were normal. Chemical and microbiological tests (included parasitological examination) on stool samples were always negative. A fecal SARS-CoV-2 PCR test resulted negative. Therefore, in view of the clinical history, in the suspicion of post-COVID subacute gastroenteritis and adenitis (of note, persistence of SARS-CoV-2 particles have been documented in the lymph nodes of children developing intussusception after or during COVID-19 [14]), we prescribed treatment with oral lactoferrin (400 mg/die) for 90 days. We also advised the mother to reintroduce gluten, milk, derivatives, and fiber in the diet of the baby. At follow-up visit, the mother told us that after about one week from the start of therapy, the symptoms had improved considerably (one daily evacuation of normal feces). She had also reintroduced into the baby’s diet all foods without problems.

## 3. Discussion

LF is a glycoprotein contained in exocrine secretions (milk, tears, saliva, bronchial and intestinal secretions) and in the secondary granules of neutrophils. Lactoferrin mainly displays many important biological properties in defense against all infectious agents. It also has an important role as an anti-inflammatory, antioxidant, and immunomodulating molecule. LF is widely used in clinical practice with evidence-based benefits in many fields such as neonatology, pulmonology, gastroenterology, allergic disorders, onco-hematology, dermatology, gynecology, and dentistry [15].

Due to its multiple biological activities, lactoferrin has been suggested as a potential therapeutic or preventive strategy for COVID-19 disease [16].

About antiviral properties, lactoferrin exhibits its action through different mechanisms, including the direct binding to the virus, the competitive mechanism, the direct inhibition of viral replication, and the stimulation of the activity of T and NK lymphocytes [17]. In particular, in SARS-CoV-2 infection, LF could play a protective role through a possible competition mechanism against the SARS-CoV-2 spike for the ACE2 receptor [18,19].

Regarding anti-inflammatory and immunomodulatory capabilities, lactoferrin can stimulate the activation of T cells and downregulate ferritin and the expression of chemotactic factors and adhesion molecules (e.g., ICAM-1, E-selectin). It can also influence the release of inflammatory cytokines such as TNF-α and IL-6, involved in the pathogenesis of MISC [20,21,22,23,24,25].

Lactoferrin exhibits its antioxidant activity by reducing tissue damage resulting from the production of free radicals and its antithrombotic activity through the modulation of the coagulative cascade and the regulation of plasminogen activity [22,23,24,25].

In this regard, in vitro studies about the antiviral activity of lactoferrin against SARS-CoV-2 infection are available in the literature [26,27,28].

Although LF is present in human exocrine secretion, it can also be supplemented to act as a nutraceutical or functional food [29]. LF can be recombinant or derived naturally from bovine or mammalian sources. It is a molecule with a high safety profile without contraindications. In fact, according to clinical studies, lactoferrin can be given in doses ranging from 100 mg to 4.5 g per day without known toxicity [30].

About in vivo studies, Campione E et al. led a clinical trial on the efficacy of an oral and intranasal liposomal bovine LF formulation. Asymptomatic and mild-to-moderate COVID-19 adult patients represented the study population. They documented a significant reduction in the mean negativization time in the treated group compared to untreated patients. They also observed a statistically significant reduction in D-dimers levels. The authors suggested that while the topical intranasal administration has a role as a protective barrier against viral infection, oral systemic administration instead reflects the anti-inflammatory and anti-thrombotic properties [31].

Rosa L. et al. reported a small retrospective study. Asymptomatic, paucisymptomatic, and moderate symptomatic COVID-19 patients, divided into a treatment and control group, represented the study population. Additionally, these authors showed that the group of patients treated with lactoferrin was negative earlier than the group of untreated patients [13].

Although there are initial studies on the role of lactoferrin in COVID-19 disease, they are limited to acute disease and adult populations. To our knowledge, the use of lactoferrin in the management of post-COVID symptoms in the pediatric population has not been described.

As previously mentioned, the exact physiopathological mechanism behind long COVID is not perfectly clear. However, currently we know that COVID-19 induces a massive release of cytokines (cytokine storm), possibly causes long-term inflammation, triggers autoimmune responses, increases the coagulation state leading to microvascular thrombosis and chronic endothelitis, and affects iron metabolism [32,33]. Moreover, studies showed that the virus is capable of persisting in some organs such as the gastrointestinal tract for months after initial infection [34].

Lactoferrin may be involved in the growth and diversification of intestinal microbiota in the gastrointestinal tract, thereby preventing the colonization and proliferation of enteric pathogens. Furthermore, it influences the proliferation and differentiation of small intestinal epithelial cells as well as the expression of epithelial digestive enzymes [35,36]. Finally, it can modulate the anti- and pro-inflammatory responses ensuring intestinal health [37].

About the use of lactoferrin in gastrointestinal disease, oral lactoferrin seems to ameliorate the course of acute gastroenteritis in children and post-antibiotic disturbances such as diarrhea [38]. There are also studies about the use of LF in newborns to prevent necrotizing enterocolitis (NEC) [39]. Moreover, a double-blind, placebo-controlled trial was conducted in pediatric patients affected by hematological malignancies receiving first-line induction chemotherapy. The authors demonstrated that oral lactoferrin supplementation was safe and that it could be useful in improving gastrointestinal disorders caused by chemotherapy [40].

In light of these considerations, specifically about the anti-inflammatory and immunomodulatory effects of LF showed in several pediatric inflammatory or infectious intestinal conditions (including our case series of two children successfully treated), along with the increasing evidence of viral persistence in the gut which, in turn, can stimulate chronic local inflammation [41], we can hypothesize that LF may be used as a potential drug for the treatment of long COVID-19 symptoms, especially the gastrointestinal ones [3], as specifically examined in randomized placebo-controlled studies. Given the long-lasting experience and safety with this medication, and the current lack of treatment options for long COVID-19 patients with gastrointestinal symptoms, we believe that such a study is biologically justified. However, it is important to clarify that long COVID-19 is a complex and probably multifactorial disease [42] and its plausible that LF may not be enough for all the phenotypes of this disease [3]. Recent case series in pediatric populations clearly highlighted the complexity of this condition and more research is needed to better understand the pathogenesis of the different phenotypes and optimal therapeutic approach [43,44,45].

## 4. Conclusions

LF displays many biological properties, including those on the gastrointestinal tract. Due to its ability and good safety profile, in light of the successful treatment of our two cases with chronic gastrointestinal symptoms after previous SARS-CoV-2 infection, we can speculate that LF might play an important role as a therapeutic strategy in long COVID symptoms, especially in gastrointestinal ones. In fact, in our reported cases, we have seen a resolution of the symptoms immediately after the beginning of the lactoferrin oral therapy. Therefore, lactoferrin may prove to be a promising therapeutic possibility, although further studies with a larger population and a placebo-controlled group are needed.

## Data Availability

Not applicable.

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
