# Peer review of "Lactoferrin as Possible Treatment for Chronic Gastrointestinal Symptoms in Children with Long COVID: Case Series and Literature Review"

_children, 2022, doi:10.3390/children9101446_

Round 1

Reviewer 1 Report

Dear authors,

An interesting  study.

Here are some aspects to improve the manuscript.

Introduction:

It should:

1. be written  the definitions of post COVID syndrome/or some descriptions of long COVID from the mentioned sources (WHO, NICE, NIH).

2.  to complete the lines 58-60 with some references about the known treatment in long COVID

Case presentation

1.      Explain the elected dose of lactoferrin and the duration of the treatment in case 1 and case 2

2.      Did the asthenia disappear after the treatment with lactoferrin in case 1?

3.      Explain where comes from the suspicion of post-covid subacute adenitis

4.      Complete lines 149-150 with more explanations available on literature

5.      I suppose lines 201-209 should be moved above line 173

6.      It is necessary to write some arguments in order to speculate the favorable effects of lactoferrin in long COVID

Conclusions

It should be mentioned the results of lactoferrin administration in the two cases

Good luck!

Author Response

thank you very much for your support in improving our paper. Please find below a point-by-point response to your comments. Changes have been highlighted in the revised version of the manuscript.

Reviewer 1

Here are some aspects to improve the manuscript.

Introduction:

It should:

  1. be written  the definitions of post COVID syndrome/or some descriptions of long COVID from the mentioned sources (WHO, NICE, NIH).

Thank you, we have added some more details on previous explanations provided in version 1.

  1. to complete the lines 58-60 with some references about the known treatment in long COVID

Thank you, we have clarified which treatments are under investigations, although evidence is very scarse

Case presentation

  1. Explain the elected dose of lactoferrin and the duration of the treatment in case 1 and case 2

Case 1: 600 mg/die for 90 days; Case 2 400 mg/die for 90 days. This is mentioned in both cases

  1. Did the asthenia disappear after the treatment with lactoferrin in case 1?

We clarified that fatigue improved more gradually in two months

  1. Explain where comes from the suspicion of post-covid subacute adenitis

Thank you, we have clarified in the case that evidence of sars-cov-2 persistence have been documented in the lymph nodes of children with intussusceptioni during or after Covid-19, and we provided the reference from a paper published in Pediatrics

  1. Complete lines 149-150 with more explanations available on literature

Thanks, those sentences are referred to references 20-25, reported in the text. I am sorry if this was referred to something else

  1. I suppose lines 201-209 should be moved above line 173

Thanks for the suggestion, we have anticipated this section

  1. It is necessary to write some arguments in order to speculate the favorable effects of lactoferrin in long COVID

Thanks a lot, we have added the following: “In light of these considerations, specifically about the anti-inflammatory and im-munomodulatory effects of Lf showed in several pediatric inflammatory or infectious intestinal conditions (including our case series of two children successfully treated), along with the increasing evidence of viral persistence in the gut which, in turn, can stimulate chronic local inflammation [41], we can hypothesized that LF may be used as a potential drug for the treatment of long- COVID 19 symptoms, especially the gas-trointestinal ones [3], ans specifically studied in randomized placebo controlled stud-ies. Given the long lasting experience and safety with this medication, and the current lack of treatment options for long Covid-19 patients with gastrointestinal symptoms, we believe that such a study is biologically justified.”

Conclusions

It should be mentioned the results of lactoferrin administration in the two cases

Thanks a lot, we have added it

Good luck!

Thank you for your support

Reviewer 2 Report

The article is well written and presents an interesting current topic of investigation.

Nevertheless, an adequate Materials & Methods section is lacking. Were the COVID-19 cases presented the only ones treated with lactoferrin at your clinic? Are all cases treated with lactoferrin at your clinic described in the article? What is the setting for the case series (private clinic, hospital...)? This was included at the beginning of Case 1 and 2 description though it should be moved to a proper Materials & Methods section.

Can the clinical improvement of both cases be supported by objective variables (such as blood exams, objective evaluation etc.)?

Moreover, some minor English language mistakes must be revised and amended.

Author Response

The article is well written and presents an interesting current topic of investigation.

thank you very much for your support in improving our paper. Please find below a point-by-point response to your comments. Changes have been highlighted in the revised version of the manuscript.

Nevertheless, an adequate Materials & Methods section is lacking. Were the COVID-19 cases presented the only ones treated with lactoferrin at your clinic? Are all cases treated with lactoferrin at your clinic described in the article? What is the setting for the case series (private clinic, hospital...)? This was included at the beginning of Case 1 and 2 description though it should be moved to a proper Materials & Methods section.

Thank you, we have added a full methods section, adding all requested changes.

Can the clinical improvement of both cases be supported by objective variables (such as blood exams, objective evaluation etc.)?

No, the only objective change was the normalization of diahrrea, and, in the case 1, the negativization of SARS-CoV-2 PCR on stool. We have clarified it.

Moreover, some minor English language mistakes must be revised and amended.

Thank you, we performed several minor corrections.